# The majority of transcripts in the squid nervous system are extensively recoded by A-to-I RNA editing

Shahar Alon[1,2], Sandra C Garrett[3], Erez Y Levanon[4], Sara Olson[3], Brenton R Graveley[3], Joshua J C Rosenthal[5†], Eli Eisenberg[2,6*†]

[1]George S Wise Faculty of Life Sciences, Department of Neurobiology, Tel Aviv University, Tel Aviv, Israel; [2]Sagol School of Neuroscience, Tel Aviv University, Tel Aviv, Israel; [3]Department of Genetics and Developmental Biology, Institute for Systems Genomics, University of Connecticut Health Center, Farmington, United States; [4]Mina and Everard Goodman Faculty of Life Sciences, Bar-Ilan University, Ramat Gan, Israel; [5]Institute of Neurobiology, University of Puerto Rico Medical Sciences Campus, San Juan, Puerto Rico; [6]Raymond and Beverly Sackler School of Physics and Astronomy, Tel Aviv University, Tel Aviv, Israel

*For correspondence: elieis@ post.tau.ac.il

†These authors contributed equally to this work

Competing interests: The authors declare that no competing interests exist.

**Abstract** RNA editing by adenosine deamination alters genetic information from the genomic blueprint. When it recodes mRNAs, it gives organisms the option to express diverse, functionally distinct, protein isoforms. All eumetazoans, from cnidarians to humans, express RNA editing enzymes. However, transcriptome-wide screens have only uncovered about 25 transcripts harboring conserved recoding RNA editing sites in mammals and several hundred recoding sites in *Drosophila*. These studies on few established models have led to the general assumption that recoding by RNA editing is extremely rare. Here we employ a novel bioinformatic approach with extensive validation to show that the squid *Doryteuthis pealeii* recodes proteins by RNA editing to an unprecedented extent. We identify 57,108 recoding sites in the nervous system, affecting the majority of the proteins studied. Recoding is tissue-dependent, and enriched in genes with neuronal and cytoskeletal functions, suggesting it plays an important role in brain physiology.

## Introduction

The central dogma of biology maintains that genetic information passes faithfully from DNA to RNA to proteins; however, with the help of tools such as alternative splicing, organisms use RNA as a canvas to modify and enrich this flow of information. RNA editing by deamination of adenosine to inosine (A-to-I) is another process used to alter genetic information (*Nishikura, 2010*). Unlike alternative splicing, which shuffles relatively large regions of RNA, editing targets single bases in order to fine-tune protein function. Because inosine is interpreted as guanosine by the cellular machinery, this process can recode codons (*Basilio et al., 1962*). A-to-I RNA editing is catalyzed by the ADAR (adenosine deaminase that acts on RNA) family of enzymes. All eumetazoans, from cnidarians to mammals, express ADARs but the extent to which they use them to recode has been explored in few representatives (*Nishikura, 2010*).

Recent advances in DNA sequencing and the supporting computational analyses have permitted transcriptome-wide screens for RNA editing events. So far, such studies have been limited to organisms with a sequenced genome (*Ramaswami et al., 2012*, *2013*). In general, these screens have looked for variation in RNA at positions that are invariant in the genome. In humans, inosine is abundant in RNA (*Paul and Bass, 1998*; *Bazak et al., 2014*), but almost all of it lies within transcribed repetitive elements in untranslated regions or introns (*Nishikura, 2010*). A compilation of recoding

**eLife digest** For living cells to create a protein, a genetic code found in its DNA must first be 'transcribed' to create a corresponding molecule of messenger RNA (mRNA). DNA and RNA are both made from smaller molecules called nucleotides that are linked together into long chains; the information in both DNA and RNA is contained in the sequence of these molecules. The mRNA nucleotides coding for proteins are 'translated' in groups of three, and most of these nucleotide triplets instruct for a specific amino acid to be added to the newly forming protein.

DNA sequences were thought to exactly correspond with the sequence of amino acids in the resulting protein. However, it is now known that processes called RNA editing can change the nucleotide sequence of the mRNA molecules after they have been transcribed from the DNA. One such editing process, called A-to-I editing, alters the 'A' nucleotide so that the translation machinery reads it as a 'G' nucleotide instead. In some—but not all—cases, this event will change, or 'recode', the amino acid encoded by this stretch of mRNA, which may change how the protein behaves. This ability to create a range of proteins from a single DNA sequence could help organisms to evolve new traits.

Evidence of amino acid recoding has only been found to a very limited extent in the few species investigated so far. There has been some evidence that suggests that recoding might occur more often, and alter more proteins, in squids and octopuses. However, this could not be confirmed as the genomes of these species have not been sequenced, and these sequences were required to investigate RNA recoding using existing techniques.

Alon et al. have now developed a new approach that allows the recoding sites to be identified in organisms whose genomes have not been sequenced. Using this technique—which compares mRNA sequences with the DNA sequence they have been transcribed from—to examine the squid nervous system revealed over 57,000 recoding sites where an 'A' nucleotide had been modified to 'G' and thereby changed the coded amino acid. Many of the identified mRNA molecules had been recoded in more than one place, and many more of these than expected changed the amino acid sequence of the protein translated from them. Alon et al. therefore suggest that RNA editing may have been crucial in the evolution of the squid's nervous system, and suggest that recoding should be considered a normal part of the process used by squids to make proteins.

sites in human transcriptomes revealed 1183 events (*Xu and Zhang, 2014*), but most were observed in only a single sample. Individual searches (*Danecek et al., 2012*; *Ramaswami et al., 2013*) uncovered only 115 (non-repetitive) recoding events, and 53 in mice; 34 recoding sites are conserved across mammals (*Pinto et al., 2014*). In *Drosophila*, an order of magnitude more recoding sites have been identified, residing in about 3% of all messages (*St Laurent et al., 2013*). Although individual editing sites are clearly essential (*Brusa et al., 1995*), these data suggest that RNA editing is not a broadly used mechanism for proteome diversification.

However, anecdotal data suggest this assumption might not apply across the animal kingdom. For example, using traditional cloning methods, scores of recoding sites have been uncovered in a small number of squid and octopus transcripts encoding potassium channels, ADARs, and ion pumps (*Garrett and Rosenthal, 2012a*). As for most organisms, there are no genomes available for cephalopods. Here we apply a novel approach for editing site detection in the absence of a sequenced genome. We use it to comprehensively identify editing sites in the squid giant axon system and other areas of the nervous system. Surprisingly, almost 60% of all mRNAs studied harbor recoding events, and most at multiple sites. These data show orders of magnitude more recoding in the squid proteome than in any other species studied to date. In squid, editing is so pervasive that the central dogma should be modified to include this process. Our results open the possibility that extensive recoding is common in many organisms, rivaling alternative splicing as a means of creating functional diversity.

## Results and discussion

To detect RNA editing sites in the squid nervous system, we generated millions of RNA and genomic DNA reads from an individual squid. Our method differed from previous approaches by using a de novo transcriptome as the point of reference instead of a genome (*Figure 1A*). The transcriptome was

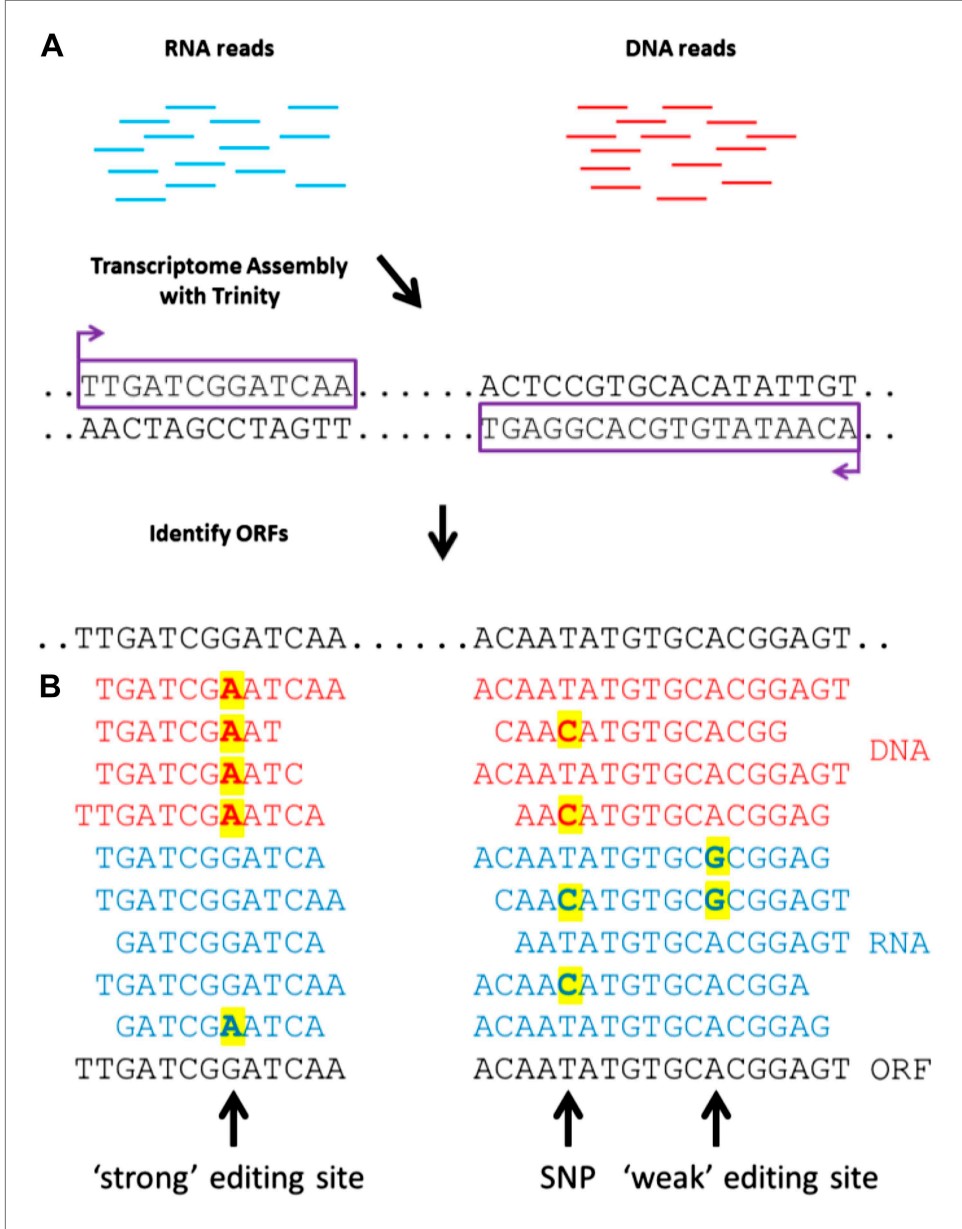

**Figure 1**. A general approach to detect RNA editing sites in organisms that lack a sequenced genome. (**A**) Squid RNA-seq data is used to create a de novo transcriptome followed by the detection of conserved ORFs. (**B**) 'Weak' and 'strong' editing sites are detected by comparing RNA and DNA reads from the same animal to the ORFs from the transcriptome. 'Weak' editing sites were detected by observing the minority of the RNA reads to differ from the consensus transcriptome nucleotide. 'Strong' editing sites, where the consensus transcriptome includes the edited nucleotide, were detected by observing all DNA reads to differ from the transcriptome nucleotide.

assembled from RNA-seq reads, and each nucleotide within it represents the consensus of many reads. If the majority of RNA reads were edited ('strong' editing sites), the transcriptome would differ from the genomic DNA and read 'G' where gDNA reads would show 'A' (the sequencing process identifies inosines as guanosines). We detected such sites by aligning DNA-seq reads to the transcriptome (*Figure 1B*). At positions where editing occurred in the minority of RNA-seq reads ('weak' editing sites), however, the transcriptome and the genomic DNA would be identical. These sites were detected by identifying variability in RNA-seq, but not DNA-seq, reads (*Figure 1B*). This general approach is applicable to all organisms that lack a sequenced genome.

We sequenced cDNA from the giant axon system (giant fiber lobe: GFL), the optic lobes (OL) and matching germline gDNA isolated from the same animal. cDNA was also sequenced from the vertical lobe (VL), buccal ganglion (BG) and the Stellate Ganglion (SG) from another animal (The SG and GFL are parts of the peripheral nervous system; all the rest are from the central nervous system). The GFL and OL RNA-seq reads were used to construct a transcriptome model (*Grabherr et al., 2011*). To focus on editing sites inside *bona fide* coding regions, we retained only transcript-fragments with open reading frames (ORFs) homologous to known proteins (*UniProt Consortium, 2014*) (*Figure 1A*) and used the editing detection procedures outlined in *Figure 1B*. Surprisingly, our pipeline identified 81,930 weak sites, and 5644 strong sites, due to A-to-G transitions (*Figure 2A*). Only 12,403 weak sites and 219 strong sites were identified for the other 11 possible types of modifications. These numbers suggest false-positive rates of 15% and 4% respectively, mainly due to transcriptome assembly problems, SNPs, somatic mutations and systematic mis-alignments. Note that these false-positive rates are considerably lower than those for similar searches for editing within human coding sequences, where a genome reference was employed (*Ramaswami et al., 2013*).

Although the number of A-to-G discrepancies was unexpectedly large, subsequent analyses support the idea that they are caused by RNA editing rather than other sources of error. First, we applied

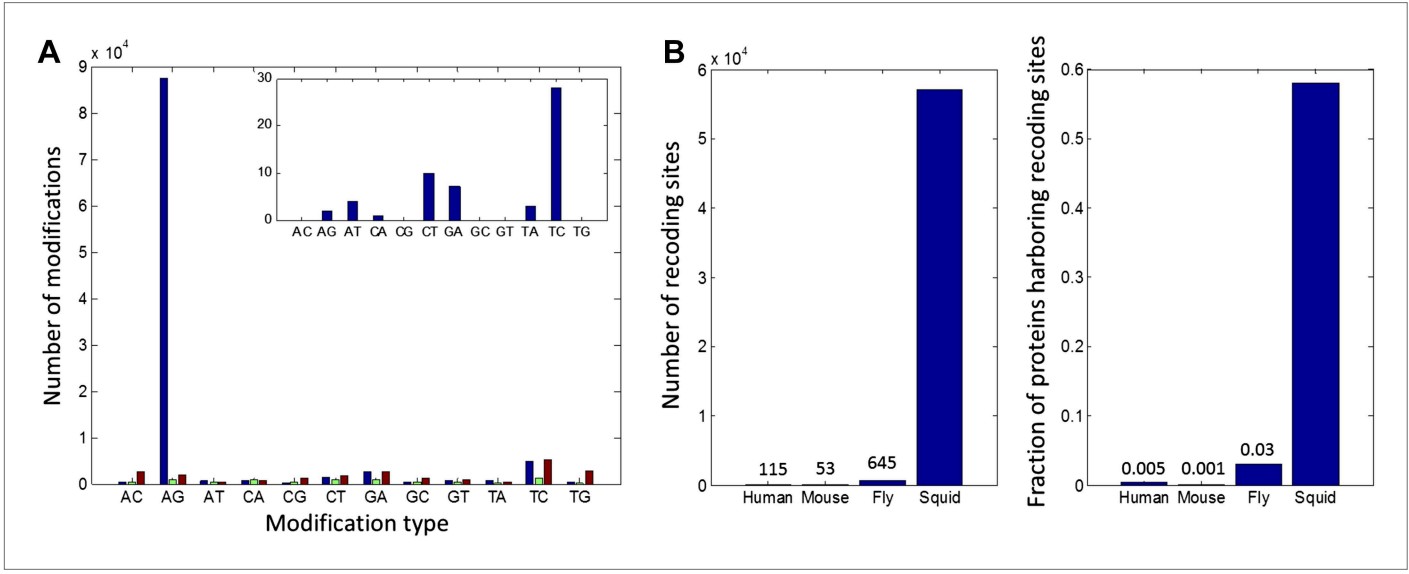

**Figure 2**. High number of RNA editing sites in squid translates into an extraordinary number of recoding events. (**A**) The number of nucleotide modifications observed in the squid nervous system for each possible substitution type (in blue, 87% of all detected modifications were A-to-G). A similar analysis of human and Rhesus macaque sequencing data (green and brown, respectively) shows low levels, and no enrichment, of A-to-I editing in coding regions, as reported previously. In the inset, the distribution of nucleotide modifications observed in squid mitochondria-encoded genes, used here as a negative control. The ADAR enzymes have no reported activity in the mitochondria and, accordingly, no A-to-G overrepresentation is observed. Also see *Figure 2—figure supplement 1*. (**B**) Scope of recoding due to RNA editing in squid, both in the total number of recoding events and the total number of genes affected, is orders of magnitude higher than human, mouse, and fly (numbers for other organisms are based on recent publications using RNA-seq datasets comparable to the one used here [*Danecek et al., 2012*; *Ramaswami et al., 2013*; *St Laurent et al., 2013*]).

The following figure supplements are available for figure 2:

**Figure supplement 1**. A-to-G modifications appear in clusters of consecutive identical mismatches and show distinctive 5′ and 3′ neighbor preferences.

**Figure supplement 2**. Hierarchical clustering reveals tissue selectivity in the modification levels of the A-to-G sites, but not in the non A-to-G sites.

**Figure supplement 3**. Validation of editing using Sanger sequencing.

**Figure supplement 4**. Quality controls for the A-to-G modifications and the non A-to-G modifications.

**Figure supplement 5**. Number of modification sites detected as a function of the amount of DNA and RNA reads.

our pipeline to similarly sized data sets from a human blood sample and from the rhesus macaque brain, each containing matching RNA and DNA sequence reads. As expected for mammals, the quantity of AG mismatches in coding regions were similar to those from non-AG mismatches, and both were quantitatively indistinguishable from the noise determined from the squid data (*Figure 2A* and *Supplementary file 1A,B*). These controls demonstrate that the enormous number of AG mismatches in the squid data is not an artefact of our analysis pipeline. Other features point to the biological origin of our AG mismatches. Similar to A-to-I editing sites in other organisms (*Morse et al., 2002*; *Levanon et al., 2004*; *Kleinberger and Eisenberg, 2010*), those identified here tend to cluster and show distinctive 5′ and 3′ neighbor preferences (*Figure 2—figure supplement 1*). In addition, hierarchical clustering of results from five tissues reveals that A-to-G modifications, but not other types, exhibit clear tissue-specificity, suggesting they do not result from genomic polymorphisms and mapping artifacts (*Figure 2—figure supplement 2*). No A-to-G overrepresentation is observed in mitochondria-encoded genes (*Figure 2A*), in agreement with the absence of ADARs, and by extension A-to-I editing, in the mitochondria. Finally, direct Sanger sequencing from a second individual confirmed editing at 40/40 A-to-G sites, and deep-sequencing validated 120/143 A-to-G sites but none of the 12 non A-to-G sites tested (*Figure 2—figure supplement 3*, *Supplementary file 1C–G*). Taken together, the overrepresentation of A-to-G modifications over all other types, the motifs surrounding the A-to-G sites, the tissue-specific modification levels, and the validation experiments, provide evidence that the majority of the A-to-G modifications are true editing events, while most non A-to-G modifications are likely technical artifacts or genomic variations (*Zaranek et al., 2010*; *Ramaswami et al., 2012*) (*Figure 2—figure supplement 4*).

Unlike with humans, the large number of A-to-I editing events translates into a large number of recoding events: Overall, 57,108 recoding events were detected in 6991/12,039 ORFs. These numbers are orders of magnitude higher than any other species studied (*Danecek et al., 2012*; *Ramaswami et al., 2013*; *St Laurent et al., 2013*; *Pinto et al., 2014*) (*Figure 2B*). Moreover, a large fraction of the proteins are recoded at multiple sites (*Figure 3A*): about 1/3 harbor ≥3 sites and 10% harbor ≥10 sites. Even when focusing only on recoding sites with editing levels >10%, about 10% of the squid proteins harbor ≥5 sites (*Figure 3A*). On the extreme end of the spectrum, the ORFs encoding α Spectrin and Piccolo have 247 and 182 recoding sites, respectively (*Figure 3B* and *Figure 3—figure supplement 1*). It should be noted that only annotated ORFs were examined in our pipeline, and the number of editing sites did not saturate with respect to the number of sequence reads (*Figure 2—figure supplement 5*). Moreover, incompleteness of the de novo transcriptome, as well as incorrect assembly of paralogs and splice variants, may cause our pipeline to miss many additional sites (*Supplementary file 1B*). Therefore there are probably many more recoding sites in the squid transcriptome.

Consistent with other organisms (*Stapleton et al., 2006*), recoding events are enriched in genes with neuronal and cytoskeletal functions (*Figure 4—figure supplement 1A* and *Supplementary file 1H*). To gain insight on the effected pathways, squid ORFs were mapped to all human KEGG pathways (*Kanehisa and Goto, 2000*). Editing has a global effect on most pathways (*Supplementary file 1I*), and those related to the nervous system are even more affected. For example, of the 27 proteins in the 'Synaptic vesicle cycle' pathway, 22 are edited and 14 heavily so (*Figure 4A*). Similarly, of the 39 proteins in the 'Axon guidance' pathway, 33 are edited and 19 heavily so. Other notable pathways are 'Regulation of actin cytoskeleton' and 'Circadian rhythm' (*Figure 4—figure supplement 1B*). By contrast, proteins in the pathways 'Ribosome' and 'RNA polymerase' are edited less than average (*Supplementary file 1I*), demonstrating that some pathways may be protected from editing. Consistently, editing levels observed in non-nervous system tissues are considerably lower (*Alon et al., 2015*).

Recently, it was suggested that RNA editing is generally not advantageous in humans (*Xu and Zhang, 2014*), as nonsynonymous events are less frequent than expected by chance (*Xu and Zhang, 2014*). Strikingly, for sites with high editing levels in squid, the opposite is true (*Figure 4B* and *Figure 4—figure supplement 2A*). Recoding events favor creation of glycine and arginine, mainly at the expense of lysine (*Figure 4—figure supplement 2B–D*). Moreover, highly edited sites within conserved domains tend to recode to amino acids that occur frequently in other species at the same position (*Figure 4—figure supplement 3*), suggesting selection towards functional substitutions and against deleterious ones.

The squid giant axon has been one of the most important models for neurophysiology. Studies using this preparation serve as a foundation for our current understanding of excitability, ion homeostasis, and axonal transport. Accordingly, we examined the extent to which RNA editing might affect

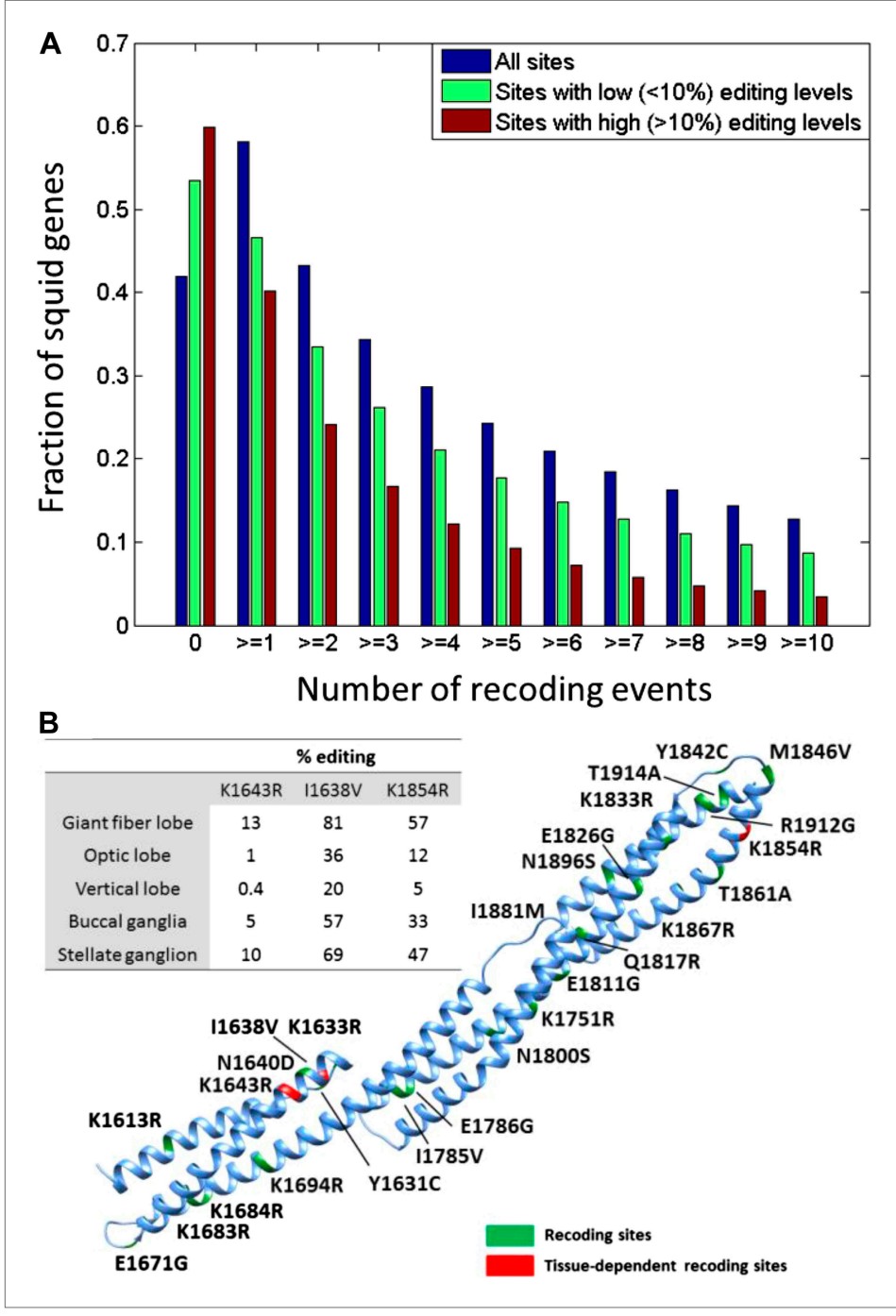

**Figure 3**. Editing often recodes multiple amino acids in the same protein. (**A**) The fraction of the squid genes that harbor multiple recoding events. About a third of the squid proteins harbor three or more recoding sites and more than 10% harbor 10 or more recoding sites. (**B**) Homology-modelling of the α Spectrin protein in which 10% of the amino acids (247/2412) are recoded by editing. Amino acids 1602 to 1918 of the squid α Spectrin protein are included in the 3-D model. Recoding sites are highlighted in green. Recoding sites with tissue-dependent levels are highlighted in red and the corresponding editing levels are indicated in the table. Also see *Figure 3—figure supplement 1*.

The following figure supplement is available for figure 3:

**Figure supplement 1**. Homology-modelling of the squid Piccolo protein in which 9% of the amino acids (182/2098) are recoded by editing.

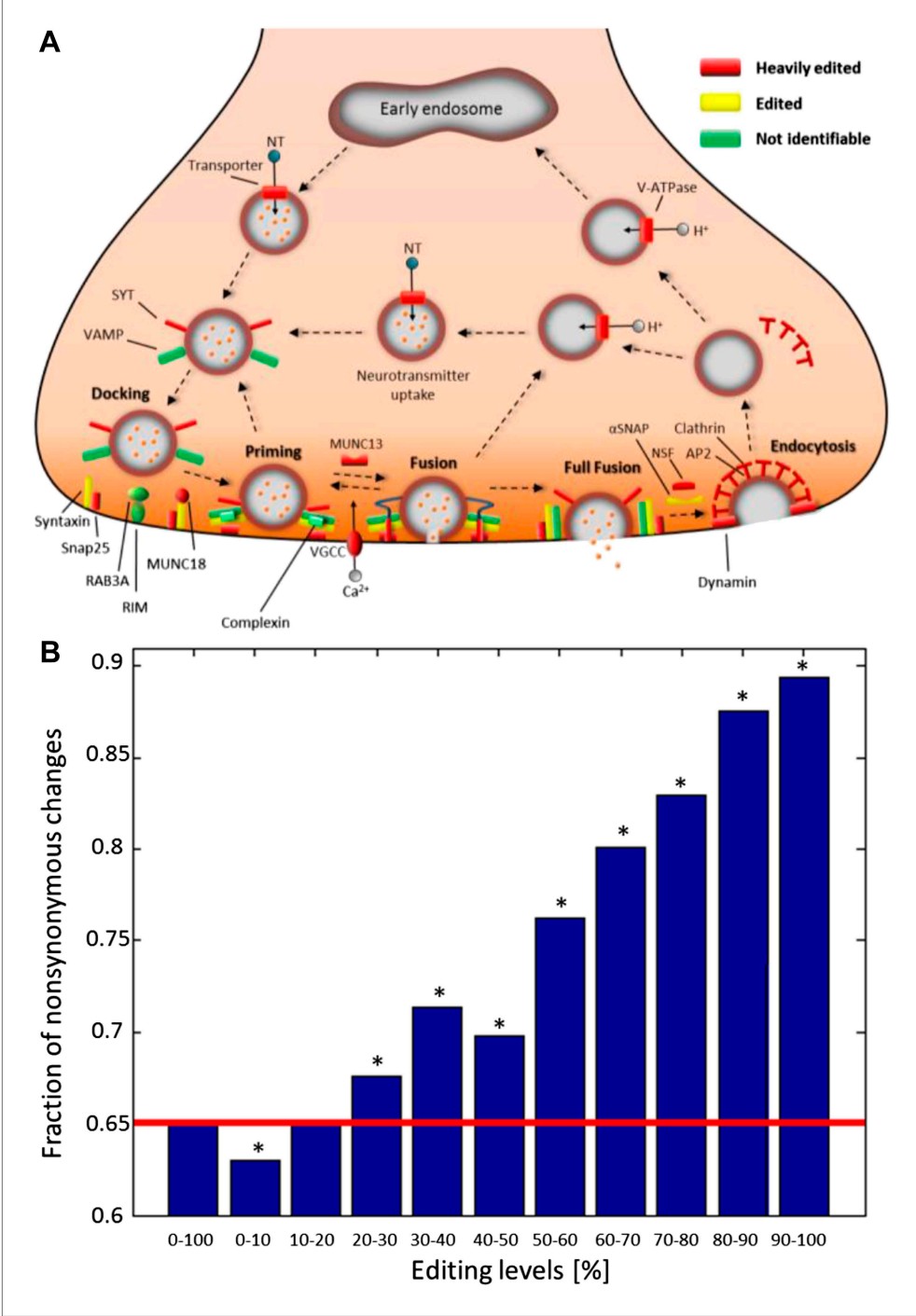

**Figure 4**. Recoding due to RNA editing affects complete molecular pathways and is likely to be more advantageous in sites with high editing levels. (**A**) All the squid proteins present in the KEGG 'Synaptic vesicle cycle' pathway are edited, and most are heavily edited. We define 'heavily edited proteins' as those for which the cumulative recoding level, that is the editing level summed over all recoding sites, exceeds unity. These are marked red, other edited proteins yellow, and proteins not identifiable in the squid transcriptome are shown in green. Also see *Figure 4—figure supplement 1*. (**B**) The fraction of nonsynonymous codon changes as a function of the editing levels, using data from the GFL and OL tissues combined. The higher the editing level, the higher the fraction of nonsynonymous codon changes. The fraction expected by chance is shown in red. A similar relationship is also true for every tissue separately (*Figure 4—figure supplement 2A*). Asterisks mark p-value <0.001 estimated using 1000 bootstrap runs.

*Figure 4. Continued on next page*

*Figure 4. Continued*

The following figure supplements are available for figure 4:

**Figure supplement 1**. Recoding events are enriched in genes with neuronal and cytoskeletal functions and globally affect molecular pathways.

**Figure supplement 2**. The fraction of nonsynonymous codon changes as a function of editing levels and the amino acid modifications due to editing.

**Figure supplement 3**. Editing tends to avoid potentially deleterious recoding events.

these processes (*Supplementary file 1J*). In total, 87 GFL ORFs that encode voltage and neurotransmitter gated ion channels, ion transporters, synaptic release machinery and molecular motors were identified in our transcriptome. In agreement with the overall editing frequency in squid, 70% harbored editing sites. Unexpectedly, however, 54% were heavily edited, many more than the 24% expected. Thus even in a background of hyper RNA editing, squid, like other organisms, preferentially edit nervous system targets.

These data suggest that editing in squid has fundamentally different underpinnings and consequences. Have squid ADARs evolved novel structures that account for the high-level editing? A past study showed that a squid ADAR2 ortholog can be expressed as two isoforms due to alternative splicing (*Palavicini et al., 2009*): one, having two double-stranded RNA (dsRNA) binding motifs, resembles vertebrate ADAR2s. A second, however, contains an 'extra' dsRNA binding motif at its N-terminus. This non-canonical isoform edits RNA more efficiently, edits more sites, and binds dsRNA with a far higher affinity (*Palavicini et al., 2009*, *2012*). Further Squid ADAR2 messages themselves contain many editing sites, leading to many subtly different isoforms. An ADAR1 isoform is also present in our transcriptome and is the focus of an ongoing study. An equally intriguing question is why squid edit to this extent? The process clearly creates tremendous protein diversity, and this may in part explain the behavioral sophistication of these complex invertebrates. A recent study showed that editing can be used for temperature adaptation in octopus (*Garrett and Rosenthal, 2012b*) and this makes sense based on the codon changes that it catalyzes (*Garrett and Rosenthal, 2012a*) (*Figure 4—figure supplement 2C–D*). In *Drosophila*, editing can respond to acute temperature changes (*Savva et al., 2012*). The large number of sites in squid suggests that editing is well positioned to respond to environmental variation. Most model organisms studied so far are mammals which are homeotherms. Future studies of more diverse species are needed to reveal the extent to which cold-blooded organisms might utilize extensive editing to respond to temperature changes and other environmental variables.

## Materials and methods

### Specimen collection and dissection

Specimens of the squid *Doryteuthis pealeii* were collected by trawl in the Vineyard Sound by the animal collection department of the Marine Biological Laboratory in Woods Hole, Massachusetts during the month of July. The giant fiber lobe (GFL) tissue of the Stellate ganglion, the optic lobe (OL) tissue and a portion of the sperm sack were manually dissected from a single adult male immersed in chilled, filtered seawater. The Buccal ganglia (BG), Stellate ganglion with the giant fiber lobe removed (SG) and Vertical lobe (VL) tissues were dissected from a second adult male individual. Tissues were also dissected from non-neuronal regions: the branchial heart, the Gill, the ventral epithelial layer on the pen, the marginal epithelial layer on the pen, the iridophore layer of the skin, and the chromatophore layer of the skin. Each of these six tissues originated from a different animal. RNA from all tissues was extracted with the RNAqueous kit (Life Technologies, Carlsbad, CA), and genomic DNA was extracted from the sperm sack using Genomic Tip Columns (Qiagen, Venlo, Limburg, The Netherlands).

### Library preparation for sequencing

The genomic DNA sequencing library was prepared using the TruSeq DNA Sample Prep kit, as described by the manufacturer (Illumina, San Diego, CA), and sequenced using three lanes of Illumina HiSeq 2000 instrument. The RNA-Seq libraries for all the samples were prepared using the TruSeq Stranded mRNA Sample Prep Kit, as described by the manufacturer (Illumina), and were sequenced

using Illumina HiSeq 2000 instrument. The GFL and OL libraries were sequenced together on a single lane, the same for the VL and BG libraries (with one unrelated library) and the SG library was sequenced on a single lane (with one unrelated library).

## Detection of editing sites in the squid nervous system

Illumina sequencing was utilized to generate ~87 million paired-end, 100 nt reads, using RNA from GFL tissue, the same number of reads using RNA from OL tissue, and ~440 million paired-end, 100 nt reads, using germline DNA. The Trinity transcript assembly tool (*Grabherr et al., 2011*) was used with the default parameters (except for 'min_kmer_cov' which was set to 2 instead of 1) to construct the genes sequences from the GFL and the OL sequencing data combined; giving 99,226 putative gene-fragments (termed 'components'). Most of the generated components were short (median of 379 bases). As Trinity attempts to identify the different transcripts (isoforms) of the components, 14,643 components were represented by more than one sequence (putative transcript-fragment). For these components, the longest sequence was selected and further used. The average length of the representing sequences for all components was 589 bases, bringing the total size of the recovered squid transcriptome to about 58 million bases.

DNA reads and RNA reads were separately aligned against the RNA components using Bowtie2 with local alignment configuration and default parameters (*Langmead and Salzberg, 2012*). Only uniquely aligned reads were used (taking only reads with the maximal 'mapping quality'). As most of the components were short, we didn't demand that both mate pairs be aligned to the same component. Instead, each read was separately analyzed. Overall, 77% (~268 million out of ~350 million) of the GFL and OL RNA reads combined were uniquely aligned to the components. However, as expected, only 3.5% of the DNA reads (~30 million out of ~880 million) were uniquely aligned to the components.

To focus on editing sites inside coding regions, and avoid repetitive elements that are prone to assembly and alignment errors, we retained only those components that were found to be significant similar (Blastx E-value<1e-6) to the Swiss-Prot proteins dataset (*UniProt Consortium, 2014*). In these components, the alignment to the homolog Swiss-Prot protein covered most of the squid sequence (63% on average). The detected ORFs were extended in both directions until a stop codon or the end of a component was reached. Overall, 12,039 ORFs, with average length of 1370 bases, were detected. These ORFs represent 8276 Swiss-Prot proteins with distinct names (two or more different ORFs may be two squid paralogs of the same Swiss-Prot protein, or fragments of the same squid gene aligned to different regions of the Swiss-Prot protein). The total size of the detected coding sequences in the squid was ~16 million bases, 28% of the total transcriptome length recovered using Trinity (the constructed squid coding sequences are given in a text file in a fasta format, *Alon et al. (2015)*, available via Dryad digital repository). The coverage of these ORFs was high; in the GFL and OL data combined, ~199 million RNA reads were aligned to these Swiss-Prot ORFs, with an average RNA coverage of 1206×. The average DNA coverage was 106× as ~21 million DNA reads, with average length of 83 bases were aligned to the coding sequences.

The above alignment data and the ORFs information were used to find the locations of all DNA-RNA mismatches inside the coding regions. In the following, bases called with quality score Q < 30 were discarded. Note, however, that Trinity consensus sequence does take into account these bases, as well as reads that might have not been uniquely aligned to the transcriptome. It is often customary to filter out reads' ends when analyzing RNA-DNA mismatches. A main reason for that is the common mismatches at reads' ends due to alignment artifacts when a splicing junction occurs near the ends. In our case, as alignment is done to the transcriptome, we did not observe any increase in AG mismatch rate near reads' ends (*Figure 2—figure supplement 4*), and thus no such filter was used. Two procedures were used to detect editing sites: (**A**) 'weak' editing sites procedure: a binomial test was applied to find the significant modifications between RNA reads and the Swiss-Prot ORFs. The binomial statistics uses the number of successes (the number of reads with a mismatch of a given type in a given position), the number of trials (the total number of RNA reads aligned to the given position) and the error probability. The probability of having a sequencing error (the error probability) was estimated using the sequence quality score. We counted only mismatches with >=30 quality score, and therefore the expected error probability was set to 0.1%. The binomial test was applied to every position inside the Swiss-Prot ORFs. The p-value for each location was corrected for multiple testing using a Benjamini-Hochberg false-discovery rate of 10%. Furthermore, in order to exclude RNA variability due to genomic

polymorphisms, we filtered out all modification sites in which any of the DNA reads aligned to the site does not agree with the transcriptome. This procedure assumes the RNA consensus in the site is identical to the gDNA reads, and thus is not suited to detect sites in which the editing appear in most of the RNA reads and therefore also in the Trinity-generated ORFs ('strong' editing sites). (**B**) 'strong' editing sites procedure: the locations in which all DNA reads showed a different base than the ORF. The probability of such sites to be not a result of editing but rather a single nucleotide polymorphism (SNP) was estimated by $(1/2)^{(\#DNA\ reads)}$ (no allele-specific expression) multiplied by the prior probability of a SNP which was taken to be 0.001. Here too, the p-value for each location was corrected for multiple testing using a Benjamini-Hochberg false-discovery rate of 10%.

Overall, 81,930 weak and 5649 strong A-to-G modification sites were detected in 7776 ORFs, and only 12,403 weak and 254 strong non A-to-G modifications sites (the modification sites and their number of supporting reads in all the tissues studied are tabulated in *Alon et al. (2015)* available via Dryad digital repository). Interestingly, 2905 of the A-to-G modification sites reside in 268 out of the 475 ORFs with only non-metazoans homologs. The number of weak A-to-G sites detected (but not strong ones) is likely to increase with RNA coverage, as we demonstrated by sampling parts of the sequencing data and re-calculating the number of A-to-G sites (*Figure 2—figure supplement 5A*). Another way to look at the dependence between the RNA coverage and the detected number of weak A-to-G sites is by recording the number of A-to-G sites detected in each ORF as a function of the ORF RNA coverage. The sorted RNA coverage was divided into ten equal bins and the mean number of A-to-G sites in each bin was calculated. As expected, higher RNA coverage is correlated with high number of A-to-G sites (*Figure 2—figure supplement 5B*). As with the RNA reads, the dependence of the detection procedures on the DNA reads was examined by sampling parts of the DNA sequencing data. Increasing the DNA coverage increased the precision of the 'weak' site detection procedure (as more SNPs are removed from the observed modifications) and strongly increased the number of detected 'strong' sites (*Figure 2—figure supplement 5C*).

We did not apply additional read-number filters, as these seem to only marginally increase accuracy while reducing the number of detected sites. However, we did exclude five strong sites for which there were no uniquely-aligned, high-quality, supporting RNA reads. This brings the number of strong A-to-G and non A-to-G modifications to 5644 and 219, respectively. The full list of weak and strong sites (*Alon et al. (2015)* available via Dryad data repository) provides the number of DNA and RNA reads per site.

We examined clusters of mismatches of the same type (several consecutive identical mismatches), revealing a high number of ORFs with A-to-G mismatch clusters (*Figure 2—figure supplement 1A*). For example, examining only ORFs with three (and above) consecutive identical mismatches, 45,199 A-to-G sites in 4265 ORFs were detected, and only 470 non A-to-G sites. Interestingly, the clusters of A-to-G sites also appear at the level of the individual reads; for example, examining only reads with four and above consecutive identical mismatches, 85% of the reads contained A-to-G modifications (*Figure 2—figure supplement 1B*). This data implies that single RNA molecules contain A-to-G clusters. We note that similar results were obtained using an algorithm which analyzes only reads with several consecutive mismatches, including reads that cannot be mapped by standard alignment tools (*Porath et al., 2014*). Using this algorithm 23,737 A-to-G modification sites were found in the squid coding regions, compared to only 2933 non A-to-G modifications. Importantly, only 728 A-to-G modifications were found in the coding regions of human using this algorithm (*Porath et al., 2014*), even though a much larger RNA-seq dataset was used (~5 billion reads of Illumina Human BodyMap 2.0), thus supporting the dramatic difference in editing between squid and other organisms.

The sequence surrounding the A-to-G modifications sites is similar for 'weak' sites and 'strong' sites, and both differ from what is expected by chance (*Figure 2—figure supplement 1C*). As expected for *bona fide* A-to-I editing sites (*Kleinberger and Eisenberg, 2010*), G is significantly underrepresented in the nucleotide before the editing site (6922 out of 87,574 sites detected) and over represented in the nucleotide that follows the editing site (38,564 out of 87,574 sites). A and T are over represented in the nucleotide before the editing site (42,475 and 25,061, respectively, out of 87,574 sites).

## Characterization of the modifications sites detected

We characterized the differences between the A-to-G modifications and all the other types of modifications (non A-to-G). One possible source of apparent modifications is sequencing errors. We thus

compared the quality score between A-to-G and non A-to-G modification sites. However, no significant difference in the distribution of the quality scores was observed (*Figure 2—figure supplement 4A*). Another technical explanation for the non A-to-G modifications can be read-end artifacts: the ends of the reads are more likely to be misaligned due to splicing, or to contain errors generated in the RNA sequencing protocol (*Ramaswami et al., 2012*). Indeed, non A-to-G modifications (unlike A-to-G ones) tend to be located in the reads-end (*Figure 2—figure supplement 4B*), suggesting a larger fraction of these sites is likely to be a result of technical artifacts. In addition, we studied the modification level distribution (the fraction of cDNA reads exhibiting the modification in a given position) for both types of modifications (*Figure 2—figure supplement 4C*). A higher fraction of non A-to-G sites show ~50% modification levels (compared to the A-to-G sites), indicating a higher fraction of missed genomic polymorphisms. Consistently, 50% of the sites with non A-to-G modification levels between 40–60% or between 90–100% recur in both the GFL and the OL tissue (coming from the same individual animal), compared to only 21% of the A-to-G modifications in the same ranges.

To find statistically significant differences in the modification levels between the GFL and the OL tissues, a binomial analysis was performed with Bonferroni-corrected p-value of 0.05 as a cutoff. Overall, 19% (16,425 out of 87,574) of the detected A-to-G modifications differ significantly between the GFL and OL tissues. Most of these sites (84%, 13,731 out of 16,425) have higher modification levels in the GFL tissue. In contrast, only 6% (783 out of 12,614) of the non A-to-G modifications are significantly different for the same two tissues, with no clear tissue preference: 45% and 55% of these sites have higher modification levels in the GFL and the OL tissue, respectively. As the same technical artifacts and genomic polymorphisms are expected to replicate in both tissues, these data increase our confidence that most of the A-to-G sites are *bona fide* editing sites. To further characterize the tissue-dependence of modifications levels, Illumina sequencing was again utilized to generate ~54, ~62 and ~42 million paired-end, 100 nt reads, using RNA from Buccal ganglia (BG), Stellate ganglion with the giant fiber lobe removed (SG), and Vertical lobe (VL), respectively, collected from a different animal. The same alignment procedure was applied to quantify the modification level at the previously described sites for each of the additional samples. Statistically significant differences in the modification levels between all the five neuronal tissues were detected using a binomial analysis with Bonferroni-corrected p-value of 0.05 as a cutoff. The sites with significantly variable A-to-G modification levels were subjected for hierarchical clustering, revealing clear tissue selectivity with higher modification levels in the GFL tissue and low levels in the VL tissue (*Figure 2—figure supplement 2A*). The same analysis for the non A-to-G modification levels demonstrates consistency between the individual animals as the modification levels in the GFL and OL tissues form one cluster, and the VL, BG and SG tissues form a second cluster, again suggesting that many of the non A-to-G modifications are due to genomic polymorphisms (*Figure 2—figure supplement 2B*).

To characterize the modification levels in non-neuronal tissues, Illumina sequencing was again utilized to generate ~23, ~23, ~19, ~26, ~19 and ~14 million paired-end, 150 nt reads, using RNA from the branchial heart, the Gill, the ventral epithelial layer on the pen, the marginal epithelial layer on the pen, the iridophore layer of the skin, and the chromatophore layer of the skin. The same alignment procedure was applied to quantify the modification level at the previously described sites for each of the additional samples, revealing considerably lower editing levels in the non-neuronal tissues (*Alon et al. (2015)*, available via Dryad data repository).

## A-to-I editing verified by Sanger sequencing and deep sequencing

Direct validation of editing was performed on a subset of the detected A-to-G sites using Sanger sequencing and deep sequencing. To reduce the chance for RNA contamination in the DNA or vice versa, primers were designed to differentially amplify the gDNA, by residing in introns, or cDNA, by spanning exon–exon boundaries. To identify intronic sequence and exon–exon junctions the following steps were performed: (a) we recorded all the cases in which the beginning or the end of the DNA read was trimmed during the local alignment against the Trinity sequences. (b) If the flanking region could be aligned against the Trinity sequences, even if aligned separately without the rest of the read, it was discarded. (c) If at least three DNA reads showed the same flanking sequence starting from the same position inside the ORFs, it was considered to be an intron fragment. This procedure revealed the positions of part of the exon–exon junctions and part of the intron sequences that correspond to the junctions. Overall, this procedure resulted in about 2100 regions, containing ~5% of all the detected editing sites, in which the gDNA and the cDNA could potentially be differentially amplified.

For the Sanger sequencing, primers were designed to differentially amplify the gDNA and cDNA of 19 ORFs (*Supplementary file 1C*). To allow better detection of editing, all the sites tested using Sanger sequencing were chosen to have >=20% modification levels (in our original GFL analysis). For the Sanger validation experiment the GFL tissue from a single animal was used (different from the animals used for the HiSeq sequencing experiments). All the sites tested (40 out of 40) were validated using Sanger sequencing (*Figure 2—figure supplement 3* and *Supplementary file 1D*).

For the deep sequencing validations, three groups of targets were tested (*Supplementary file 1E,F*): (a) twenty 'interesting' genes, that is, squid components with homology to genes known to be implicated in human diseases or in other important pathways, selected from the dataset of ~2100 regions described above. (b) 20 regions randomly-selected from the above dataset of ~2100 regions. (c) 20 regions randomly-selected from the set of putatively edited regions for which we could not design unique gDNA primers, and thus gDNA and the cDNA could not be differentially amplified. For this group the same primers were used to amplify the gDNA and the cDNA. As with the Sanger validation experiment, the deep sequencing validations were done using GFL tissue from a single animal (different from the animals used for the HiSeq and Sanger sequencing experiments). All primer sets were designed with an overhang so that sequencing and indexing primers could be added to the amplicons in a nested PCR reaction. Samples from gDNA and cDNA were distinguished by unique sequencing indexes. After nested PCR, amplicons were pooled, purified by Ampure XP (Beckman Coulter, Danvers, MA), and sequenced on an Illumina MiSeq instrument. All the cDNA targets were amplified, but only 49 of the 60 gDNA targets amplified well enough for analysis (*Supplementary file 1E,F*). For the other eleven targets, the presence of undetected introns could have disrupted amplification. The DNA and RNA reads were analyzed using the same detection procedure described above (*Figure 1B*), with the sole exception that sequence variations below 0.1% (the expected sequencing error rate) were allowed in the DNA reads (as mandated by the much larger DNA coverage in this validation study). Altogether, 84% (120 out of 143) of the A-to-G sites examined were validated (*Supplementary file 1F,G*). In contrast, none of the 12 non A-to-G sites examined were validated. Moreover, 170 additional A-to-G sites (86% of all novel detected sites, a similar fraction to the HiSeq data used in the original detection, *Figure 2A*) were identified, more than doubling the number of A-to-G sites detected. Similar results were observed for the three groups examined. Finally, for the validated A-to-G sites, high correlation was obtained between the editing levels (the fraction of cDNA reads exhibiting the editing in a given position) measured using the HiSeq data and the MiSeq data (Pearson's r of 0.86, p-value = 1e-35) (*Supplementary file 1G*).

## The effect of the A-to-I editing on the proteome

To demonstrate the extent of massive recoding on two examples, homology-modelling of the squid proteins α Spectrin and Piccolo was performed with SWISS-MODEL using default parameters (*Biasini et al., 2014*) and visualized using UCSF Chimera package http://www.cgl.ucsf.edu/chimera (*Figure 3B* and *Figure 3—figure supplement 1*).

To find if the recoding events are enriched in genes with specific functions, we have calculated the cumulative recoding level, that is, the editing level summed over all recoding sites within each squid ORF. This gives a single score representing the extent of recoding in the whole protein. The squid ORFs list was ranked using the cumulative recoding level and all the Swiss-Prot annotations were converted to human Swiss-Prot annotations (when possible) for consistency. The GO analysis tool GOrilla was used to find enriched GO annotations in the ranked list (*Eden et al., 2009*). In order to control for possible detection bias in highly expressed genes, the same list was ranked using the gene expression level (FPKM) and was also analyzed using GOrilla. As expected, the enriched GO annotations in the control list (that is, genes ranked by expression levels) were mainly connected to the ribosome (translational elongation, structural constituent of ribosome, RNA binding and so on). In contrast, the list ranked by the cumulative recoding level gave enriched GO annotations which are mainly connected to neuronal and cytoskeleton functions (*Figure 4—figure supplement 1A* and *Supplementary file 1H*). Trying to detect enriched GO annotations at the bottom of the list (ranked by the cumulative recoding level) did not produce any significant results.

The extensive recoding due to RNA editing can affect many molecular pathways. In order to appreciate the extent of this phenomenon, squid ORFs were mapped to all human KEGG pathways (*Kanehisa and Goto, 2000*), revealing that RNA editing has a global effect on the majority of the squid pathways (*Supplementary file 1I*). In this analysis, the homologs to the human proteins can be: (a) 'heavily

edited', defined as proteins for which the cumulative recoding level, that is the editing level summed over all recoding sites, exceeds unity, (b) 'edited', if the protein has at least one recoding site, (c) not edited, or (d) not identifiable in the squid transcriptome. Editing levels were calculated using data from the GFL and OL tissues combined. On average, 74% and 22% of the identifiable proteins in each pathway are edited or heavily edited, respectively. Pathways related to the nervous system are even more extensively edited: 75% and 35% of the identifiable proteins in these pathways are edited and heavily edited, respectively. On the flip side, in the pathways 'Ribosome' and 'RNA polymerase' only 50% and 33% of the identifiable proteins are edited and 0% and 4% of the identifiable proteins are heavily edited, respectively (*Supplementary file 1I*), demonstrating that some crucial pathways are protected from editing. We have also specifically examined the extent of recoding in squid ORFs that encode voltage and neurotransmitter gated ion channels, ion transporters, synaptic release machinery and molecular motors. Overall, we examined 87 ORFs that are homologous to the following proteins: Voltage-gated potassium channel alpha subunit, Voltage-dependent sodium channel alpha subunit, Voltage-dependent calcium channels, Ionotropic glutamate receptors, Synaptotagmin, Synaptobrevin, Syntaxin, Synapsin, Snap 25, Sodium/potassium-transporting ATPase alpha subunit, Sodium/calcium exchanger (SLC8), Sodium bicarbonate exchanger (SLC4), Sodium/hydrogen exchanger (SLC9), Sodium/potassium/calcium exchanger (SLC24), Dynein, and Kinesin (*Supplementary file 1J*). In the GFL tissue, 47 out of the 87 proteins (54%) are heavily edited, more than twofold higher than expected by chance (p-value<1e-6). In fact, in all the examined neuronal tissues, this group of 87 proteins is heavily edited, significantly higher than expected by chance (*Supplementary file 1J*).

An important question is whether editing in the squid ORFs tends to avoid recoding by preferring synonymous modifications or, alternatively, tends to create more recoding sites than expected by chance. In squid, the expected fraction of nonsynonymous changes is 0.65, estimated by random changes of adenosines in the ORFs, accounting for the observed local sequence preference of the editing sites (*Figure 2—figure supplement 1C*), as follows: (a) we examined the sequence surrounding sites detected as edited (the base preceding and the following the site), and counted the number of times each one of the 16 possible combinations of upstream and downstream nucleotide appears. These numbers were normalized by the number of times each combination appears for all the A bases in all the ORFs, to produce the observed probability of being targeted by editing given a certain combination of 5' and 3' nucleotides. (b) all the locations inside the ORFs were screened in a random order until an A base was encountered. (c) a random number was generated, if it was below the normalized probability corresponding to the sequence surrounding the site in question, this site was selected for further use. (d) the potential effect on the codon due to changing the A base to G was examined, in particular, whether the change is nonsynonymous or synonymous. (e) steps (b–d) were repeated until the number of A bases changed matched the observed number of editing sites. The described randomization procedure was also used to calculate the expected codon changes due to editing (*Figure 4—figure supplement 2B*). We found that the higher the editing levels, the higher the fraction of nonsynonymous changes, and for editing levels >20% the nonsynonymous fraction is significantly higher than the expected fraction of 0.65 (*Figure 4B* and *Figure 4—figure supplement 2A*).

A recoding event which creates an amino acid rarely found in homologous proteins may indicate that the editing is deleterious. Therefore, each squid ORF was aligned against the conserved domains in the Conserved Domain Database (CDD) using DELTA-BLAST (*Boratyn et al., 2012*; *Marchler-Bauer et al., 2013*). DELTA-BLAST calculates the position-specific score matrices (PSSM) for each possible amino acid substitution. Positive scores indicate that a certain amino acid substitution occurs more frequently in the alignment against the conserved domains than expected by chance, while negative scores indicate that the substitution occurs less frequently than expected. The substitution score of each editing event was recorded as well as the editing level in the same position, using data from the GFL and OL tissues combined. As a control, the editing levels in all the sites with the same recoding type were randomly shuffled. This accounts for the fact that editing creates specific amino acid changes (*Figure 4—figure supplement 2B–D*), and in turn, these specific changes may be correlated with average editing levels and with substitution scores. Thus, the shuffled dataset preserves both the distribution of recoding types and the distribution of editing levels for each recoding type. This analysis revealed that the average editing levels in sites with large negative substitution scores, which may be deleterious, are significantly lower than what is expected by chance (*Figure 4—figure supplement 3A*). The distribution of recoding sites for each substitution score was calculated and compared

with random changes that preserve the sequence preference and the total number of editing events (*Figure 2—figure supplement 1C*). Consistent with the above finding, there are significantly less recoding sites with large negative substitution score than expected by chance (*Figure 4—figure supplement 3B*). Finally, on average, the higher the editing level in a given site, the higher the average substitution score, above what is expected by chance (*Figure 4—figure supplement 3C*). Thus, this analysis indicates that recoding by editing tends to avoid potentially deleterious sites and that editing sites with high editing levels might be more important functionally than editing sites with low editing levels.

## Applying our detection procedure for human and rhesus macaque sequencing data

We searched publicly available datasets and found two datasets of matched DNA- and RNA-seq data which are comparable in size and read-lengths to our squid data: (i) Human RNA-seq and DNA-seq data of blood samples (*Chen et al., 2012*) (ii) Macaque RNA-seq and DNA-seq brain data (*Chen et al., 2014*). The same pipeline as described above was applied to the data, with one single exception: while comparing the transcriptome model with SwissProt we used only non-vertebrate SwissProt sequences, in order to mimic the situation for squid in which there are no SwissProt entries from closely related species. The analysis of the human and macaque data was done for two purposes: (**A**) to show that the enormous number of AG mismatches in the squid data is real and not an artefact of the analysis pipeline, and (**B**) to demonstrate that our pipeline could identify RNA editing sites established by previous studies. The results of the analysis are summarized in *Figure 2A* and *Supplementary file 1A*. For non-AG mismatches, we obtained roughly the same numbers as those for squid. However, as expected, the AG mismatches in the human and macaque samples did not show any enrichment, and their abundance (in coding regions) was similar to those for other mismatches. In other words, the sensitivity of our method allows the detection of the super-strong editing signal of squid, but cannot separate the rare editing sites in mammals (in coding regions) from noise. Note that the task of detecting recoding sites in mammals is highly non-trivial even when one takes advantage of all information available, including the accurate genome sequence. The best efforts, so far, have yielded enrichment of AG mismatches, but still most detected sites are non-AG (i.e., most-probably, false-positives) (*Ramaswami et al., 2013*).

In order to assess the precision of our method in recalling true editing sites we have looked at the validated editing sites found in the two studies above, and checked whether they have been picked up by our algorithm as well. The full data is presented in *Supplementary file 1B*. Overall, about half of the sites were picked up by our pipeline. The sites that were not identified, mostly resided in regions poorly described by our Trinity transcriptome assembly (some were just very weakly edited). Thus, one may conclude that the recall rate of our method is lower than 0.5, and the true extent of squid recoding is even much larger than we report.

## Data Deposit

The data reported in this paper was deposited to the Sequence Read Archive (SRA), under accession SRP044717. The constructed squid coding sequences and all the A-to-G modification sites detected in the coding regions of the squid are available via Dryad digital repository (*Alon et al., 2015*; http://dx.doi.org/10.5061/dryad.2hv7d).

## Acknowledgements

We thank Hagit Porath and Dr Michal Barak for technical help and fruitful discussions.

## Additional information

### Funding

| Funder | Grant reference number | Author |
|---|---|---|
| European Research Council | grant no. 311257 | Erez Y Levanon |
| Israel Science Foundation | [379/12] | Eli Eisenberg |
| National Institutes of Health | R01 NS064259NIH | Joshua J C Rosenthal |

| Funder | Grant reference number | Author |
|---|---|---|
| National Science Foundation | HRD-1137725 | Joshua J C Rosenthal |
| Marine Biological Laboratory | | Joshua J C Rosenthal |
| John and Donna Krenicki Endowment Fund | | Brenton R Graveley |
| Clore Israel Foundation | Graduate Student Fellowship | Shahar Alon |
| United States–Israel Binational Science Foundation | 2013094 | Joshua J C Rosenthal, Eli Eisenberg |

The funders had no role in study design, data collection and interpretation, or the decision to submit the work for publication.

## Author contributions

SA, conceived the study, analyzed the data (with input from JJCR and EYL), wrote the manuscript with input from all authors; SCG, performed experiments (with input from SA and EE); EYL, conceived the study; SO, performed experiments (with input from SA and EE); BRG, conceived the study, performed experiments (with input from SA and EE); JJCR, conceived the study, performed experiments (with input from SA and EE), analyzed the data (with input from JJCR and EYL), wrote the manuscript with input from all authors; EE, conceived the study, EE analyzed the data (with input from JJCR and EYL), wrote the manuscript with input from all authors

## Ethics

Animal experimentation: Animal experimentation was conducted in accordance to the guidelines of the Marine Biological Laboratory in Woods Hole, Massachusetts.

# Additional files

## Supplementary file

• Supplementary file 1. Statistics, Primers and Data. (**A**) Statistics of the human and macaque datasets analyzed using our detection procedure. (**B**) About half (15/33) of the validated editing sites in human blood sample and Macaque brain sample were identified by our detection procedure. (**C**) Primers used for the Sanger validation experiments in 19 ORFs. (**D**) 40 editing sites validated using Sanger sequencing. (**E**) Primers used for the MiSeq validation experiments in 60 ORFs. (**F**) Summary of the MiSeq validation experiments. (**G**) 143 A-to-G modification sites detected using the HiSeq data and were either validated (120 sites) or not validated (23 sites) using the MiSeq data. (**H**) The Gene ontology (GO) terms enriched in a list of squid ORFs ranked by the cumulative recoding level, that is the editing level summed over all recoding sites. (**I**) Squid ORFs mapped to all human KEGG pathways. (**J**) Recoding in 87 squid ORFs that encode voltage and neurotransmitter gated ion channels, ion transporters, synaptic release machinery and molecular motors.

## Major datasets

The following datasets were generated:

| Author(s) | Year | Dataset title | Dataset ID and/or URL | Database, license, and accessibility information |
|---|---|---|---|---|
| Alon Shahar, Garrett Sandra C, Levanon Erez Y, Olson Sara, Graveley Brenton R, Rosenthal Joshua JC and Eisenberg Eli | 2015 | Data from: The majority of transcripts in the squid nervous system are extensively recoded by A-to-I RNA editing | http://dx.doi.org/10.5061/dryad.2hv7d | Available at Dryad Digital Repository under a CC0 Public Domain Dedication. |
| Alon Shahar, Garrett Sandra C, Levanon Erez Y, Olson Sara, Graveley Brenton R, Rosenthal Joshua JC and Eisenberg Eli | 2014 | Raw sequencing data | SRP044717 | Publicly available at the NCBI Sequence Read Archive (http://www.ncbi.nlm.nih.gov/sra). |

The following previously published datasets were used:

| Author(s) | Year | Dataset title | Dataset ID and/or URL | Database, license, and accessibility information |
|---|---|---|---|---|
| Chen R, Mias GI, Li-Pook-Than J, Jiang L, Lam HY, Chen R, Miriami E, Karczewski KJ, Hariharan M, Dewey FE, Cheng Y, Clark MJ, Im H, Habegger L, Balasubramanian S, O'Huallachain M, Dudley JT, Hillenmeyer S, Haraksingh R, Sharon D, Euskirchen G, Lacroute P, Bettinger K, Boyle AP, Kasowski M, Grubert F, Seki S, Garcia M, Whirl-Carrillo M, Gallardo M, Blasco MA, Greenberg PL, Snyder P, Klein TE, Altman RB, Butte AJ, Ashley EA, Gerstein M, Nadeau KC, Tang H, Snyder M | 2012 | Human RNA-seq and DNA-seq data of blood samples | SRP008054.4, SRP008976 | Publicly available at the NCBI Sequence Read Archive (http://www.ncbi.nlm.nih.gov/sra). |
| Chen JY, Peng Z, Zhang R, Yang XZ, Tan BC, Fang H, Liu CJ, Shi M, Ye ZQ, Zhang YE, Deng M, Zhang X, Li CY | 2014 | Macaque RNA-seq and DNA-seq brain data | SRP039366, SRP009818 | Publicly available at the NCBI Sequence Read Archive (http://www.ncbi.nlm.nih.gov/sra). |

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
