## [Decision Letter]

Thank you for sending your work entitled “The majority of transcripts in the
squid nervous system are extensively recoded by A-to-I RNA editing” for
consideration at *eLife*. Your article has been favorably evaluated by
Chris Ponting (Senior editor) and 2 reviewers, one of whom is a member of our Board of
Reviewing Editors.

The Reviewing editor and the other reviewer discussed their comments before we reached
this decision, and the Reviewing editor has assembled the following comments to help you
prepare a revised submission.

The paper by Alon et al. is a well-performed, concise study that shows extensive RNA
editing in the squid genome. Since the extent of editing is several orders of magnitude
larger than that reported in the human genome, and actually in any other metazoan
species, the results are obviously of biological relevance, and therefore appropriate
for *eLife*. The manuscript also describes a novel bioinformatic method
to identify editing sites. Overall, the manuscript is well written and both Methods and
Results are clearly explained.

Despite being unexpected, the results appear to be quite robust: the control in primates
shows that they are not an artifact from novel pipeline employed to predict editing
sites in absence of assembled genome sequences. The strong bias to AG mismatches, the
clustering pattern in genes, the neighbor preferences, the tissue specificity
(irrespective of biological origin), the resulting recoding events towards the common
amino-acid, etc., all these strongly support the editing sites found by the authors.

1) The de novo transcriptome assembly is a very trivial computational issue. Many false
positives are expected at least in complex mammalian transcriptomes. Paralogs could
affect the reconstruction of real isoforms leading to a sort of chimeric transcripts. In
addition, alternative splicing may complicate transcript reconstruction. Are there
estimations about the impact of alternative splicing and paralogs in squid? Any impact
of this on the results should be discussed in the text. Also, the text should clarify
that this is not a completely de novo method since genomic sequences are generated.

2) The strategy is biased towards the RNA editing prediction in protein coding regions
(CDS). Can RNA editing events be detected also in non-CDS regions by the method? If not,
this should be clarified in the text. Related to this, evidence of RNA editing in
repetitive regions in squid could potentially be interesting, probably revealing an
opposite trend than mammals.

3) Regarding methodology, can the statistical binomial test detect any significant
change in the non-AG positions? If yes, how do you explain this finding?

The average RNA and DNA coverage is high but regarding RNA editing candidates, are there
filters to exclude low covered sites? What is the minimal coverage for RNA and DNA?

Did you apply filters to RNA and DNA reads? I mean reads with low quality and positions
at read ends.

4) Have the authors considered the possibility that their results arise from somatic
genomic editing, rather than RNA editing? While for the human and macaque control, the
RNA and DNA samples are from the same tissues, in the case of squid, RNA samples are
from the tissues from the nervous system, while DNA is from the sperm sack. To
unequivocally conclude that the observations are indeed from RNA editing, I guess that
DNA and RNA need to be from the same biological source. Maybe the investigation of the
distribution of the relative proportion of reads supporting and not supporting the edit
could help here.

5) Related to the above, the authors used RNA only from tissues from the nervous system.
Therefore, it is not possible to assess whether the phenomenon observed is
characteristic of this system, or it is actually systemic in the entire organism. I
think that sequencing RNA from some other non-nervous tissue could help to distinguish
between the two hypotheses.

6) Regarding the characterization of RNA editing events, events tend to be tissue
specific. Are there events showing tissue specific levels? That is, cases in which the
gene locus in expressed at the same level in all tissues but editing levels are
different.

7) It is a little bit disappointing that there is limited investigation in the potential
mechanisms behind the extensive editing observed. The authors could have at least
investigated ADAR with some additional detail. The RNA (and DNA sequence) helps to
delineate the ADAR sequence, and the RNA reads to estimate expression levels. Are there
multiple copies of ADAR in the squid genome? Is ADAR expressed at comparatively higher
expression levels than in organism with low editing levels (they can use the mouse and
human samples to make this comparison? Has the ADAR sequence in squid diverged faster
than expected? In specific domains? All these questions are quite simple to answer.

8) The authors also provide an adaptive explanation to the high levels of editing
observed in the squid genome, and hypothesize that, in contrast to current assumptions,
that extensive editing is common as a way to cope with temperature adaption, except in
mammals that, as homeotherms, would not require such a process. This is, by the way,
reminiscent of the isochore theory by Bernardi that would separate homeotherm
vertebrates from “cold-blooded” (poikilotherm) vertebrates (to which, by
the way, the authors may want to cite). If the authors were correct that would indeed be
a quite relevant result. They could easily employ their pipeline in available vertebrate
RNAseq data (for instance, http://www.sciencemag.org/content/338/6114/1587.full) to test this
hypothesis.

---

## [Author Response]

Before we address your report point-by-point, we would like to reiterate an important
issue that is touched upon in many of your queries: our editing detection pipeline,
relying on a de novo transcriptome assembly, is inferior to established methods which
align RNA reads to a genomic reference. All tools for de novo transcript assembly have
limited accuracy, particularly when they try to predict close paralogs and splice
variants from short cDNA reads. These shortcomings may lead to false-positives in our
pipeline, which add to errors caused by the more familiar somatic mutations, SNPs, and
alignment errors that are common to all editing detection schemes. As is the case for
all editing detection pipelines, the challenge is to minimize noise level to the extent
that true editing sites are not buried in the noise. Here, we are able to apply our
pipeline to squid despite the above described errors because the editing signal is
enormous.

We are confident that the vast majority of A-to-G discrepancies that we have identified
reflect true RNA editing events for many reasons. First, all of the sources of error
outlined above are not expected to be biased towards A-to-G mismatches. Even if they
were, they should not be biased towards the expressed strand of the DNA (i.e. one should
expect equal numbers of A-to-G and T-to-C mismatches on the expressed strand if they
were true errors). Accordingly, we conclude that the level of non-AG mismatches
detected, which we used to estimate the false-discovery rate, already includes the sum
of all of the above types of error (and any additional ones of unknown source). The
dramatic overrepresentation of A-to-G mismatches over non-AG ones (including T-to-C) is
therefore strong evidence that most of the detected A-to-G modifications are due to bona
fide ADAR editing. Our conclusions are further supported by the fact that edited
adenosines are surrounded by ADAR’s preferred nucleotide motif, that, as with
other organisms, editing sites are clustered, and through direct validation of a
randomly selected subset of the sites. That being said, we do expect a false discovery
rate of 10-15% in our list of ∼90,000 predicted sites, and these thousands of
false-positives are probably due to inaccurate assembly, SNPs, somatic mutations and
misalignments.

*1) The de novo transcriptome assembly is a very trivial computational issue.
Many false positives are expected at least in complex mammalian transcriptomes.
Paralogs could affect the reconstruction of real isoforms leading to a sort of
chimeric transcripts. In addition, alternative splicing may complicate transcript
reconstruction. Are there estimations about the impact of alternative splicing and
paralogs in squid? Any impact of this on the results should be discussed in the
text*.

We completely agree that de novo assembly of complex transcriptomes is bound to result
in many misidentifications of paralogs and splice variants. This may impact our results
by leading to some false detection of “editing“ sites as well as by
missing true sites. As mentioned above, the total false discovery rate can be estimated
by non-AG mismatches, and is rather low. In fact, it is much lower than genome-aware
methods in model organisms, which identify only ∼40% A-to-G mismatches in coding
sequences (see e.g. [Supplementary-material SD1-data] of Ramaswami et al., 2013). The reason is not that our pipeline is
better than the genome-aware methods; it's just that the squid recoding signal is
very strong.

It is difficult to estimate the number of false positives due to mis-assembly, but one
may use the total number of T-to-C mismatches (∼5000) to approximate the total
false discovery rate, which is an upper bound for false-discovery rate due to
mis-assembly. As discussed in the text, it seems that SNPs are a major contributor to
false discoveries, so not all of these ∼5000 sites are due to mis-assembly.
Moreover, by applying our pipeline to primate datasets of similar size we identify
≤1000 sites for each mismatch type. Thus one might roughly estimate the number of
false positives due to mis-assembly at a few thousands at most.

As for false-negatives, an incomplete transcriptome may lead to missing true recoding
sites. The scope of this may be demonstrated by our primate analysis (Table B), where
the major reason for not identifying about half of the known recoding sites is the
absence of these sites in the de novo transcriptome, or lack of resolution of different
splice variants (see Table B and footnotes). Accordingly, the full recoding repertoire
in the squid may be roughly twice what we report here.

In summary, we estimate the number of false-positives due to mis-assembly to be up to a
few thousands. This source of error is included in our false discovery rate, and is far
lower than the true signal. As for false-negatives, we can only extrapolate from the
primate data, and roughly estimate their number to be sizable, making the full recoding
repertoire of the squid up to twice the size reported here.

We have amended the text to the following: “This suggests false-positive rates of
15% and 4% respectively, mainly due to transcriptome assembly problems, SNPs, somatic
mutations and systematic mis-alignments. Note that these false-positive rates are
considerably lower than those for similar searches for editing within human coding
sequences, where a genome reference was employed.”

And also: “Moreover, incompleteness of the de novo transcriptome, as well as
incorrect assembly of paralogs and splice variants, may cause our pipeline to miss many
additional sites (Table B).”

*Also, the text should clarify that this is not a completely de novo method since
genomic sequences are generated*.

Our transcriptome assembly does not use the DNA-reads in any way, and thus is indeed
'de novo'. Editing detection does use DNA-reads, and accordingly we do not
refer to our pipeline as a de novo method for editing detection.

*2) The strategy is biased towards the RNA editing prediction in protein coding
regions (CDS). Can RNA editing events be detected also in non-CDS regions by the
method? If not, this should be clarified in the text. Related to this, evidence of
RNA editing in repetitive regions in squid could potentially be interesting, probably
revealing an opposite trend than mammals*.

The question of editing in squid repeats is indeed interesting; however without a
fully-assembled genome available, it is difficult to attack at this point. The main
reason is that non-coding regions in the squid genome tend to be rich in repeats that
dramatically affect the quality of the de novo transcriptome analysis. Thus, while the
de novo transcriptome does include some non-coding sequences, and our method may, in
principle, apply (hence it is not essentially biased towards CDS), we chose to focus
only at CDS in order to improve our signal-to-noise ratio.

We have amended the text to read: “To focus on editing sites inside coding
regions, and avoid repetitive elements that are prone to assembly and alignment errors,
we retained only those components that were found to be significant similar (Blastx
E-value<1e-6) to the Swiss-Prot proteins dataset.”

3) Regarding methodology, can the statistical binomial test detect any
significant change in the non-AG positions? If yes, how do you explain this
finding?

The statistical test is designed to filter out sequencing errors. Thus, many mismatches
that are not editing sites pass these tests, such as (i) SNPs not identified due to
insufficient DNA-seq reads coverage (ii) somatic mutations (iii) mis-alignments (of DNA
and/or RNA reads) and (iv) errors in the underlying transcriptome, due to the issues
mentioned in point #1. In addition, even for the sequencing errors, the
multiple-correction scheme we use (Benjamini-Hochberg) does not guarantee zero
false-positives.

All of the above lead to the thousands of non-AG mismatches observed. As mentioned
above, the data presented in Figure 2—figure supplement 2 suggests that SNPs are the major source of error.

The average RNA and DNA coverage is high but regarding RNA editing candidates,
are there filters to exclude low covered sites? What is the minimal coverage for RNA
and DNA?

We thank the referees for raising this important issue that was not properly discussed
in our manuscript. In order to minimize the number of arbitrary parameters in our
pipeline, we did not apply any additional DNA and RNA-coverage filter, except for the
implicit requirement for enough read coverage to attain significance in the statistical
tests. Such additional filters do improve the quality of the results (in terms of
signal-to-noise), but we still feel they are not necessary (except for one minor
modification, see below) as we now explain:

Weak sites detection is based on a binomial test applied to the RNA reads mapped to the
transcriptome. The minimal number of RNA reads to significance in this case is two
reads, both of which are mismatched. There are only two such cases among the 81930 weak
sites.

One may wonder how come a site with 2 reads reading “G” could have an
“A” in the consensus (as is the case for weak sites). The answer is that
while our alignments and statistical tests take into account only uniquely aligned reads
and quality scores Q≥30, Trinity does not apply these criteria. We chose not to
fiddle with Trinity transcriptome assembly and use it as is, and thus there are quite a
few cases where the consensus nucleotide of Trinity differs from the majority of the
reads we considered in our downstream analysis.

Another possible filter is the number of DNA reads. As mentioned above we did not apply
any filter of this number. In fact, among our 81930 weak sites there are 1108 sites with
no DNA reads at all, and 1148, 1581, 2110 and 2753 sites with 1,2,3,4 supporting DNA
reads, respectively. At first, one could have thought that these sites should be
discarded, as they might very well be just genomic SNPs. However, looking at the non-AG
mismatches, and repeating our analysis with the additional filter of
minimum-N-DNA-reads, we get the following results:Minimum DNA reads#AG sites%AG sites#TC (2nd most abundant mismatch)0 (No filter)8193086.9%479318075087.5%462027941087.9%446037773188.3%427947539088.6%406457250988.9%3879

*Note that the counts of AG sites are not exactly the same as the numbers quoted
above for the current dataset. This is because each change of the filters influences the
downstream false discovery rate significance calculation.

So, while adding the DNA reads filter does marginally improve accuracy, the number of AG
sites lost (roughly 1800 sites per one additional DNA read required) is an order of
magnitude larger than the false-positive sites weeded out (quantified by the number of
TC sites). We therefore choose not to imply the additional filter, and keep the
thousands of sites with low DNA reads coverage.

Notably, eight of the 143 sites tested in our MiSeq validation were taken from this
group. All of these eight sites were confirmed to be editing sites (see Table G). In
addition, most of these sites show evidence of editing in the additional tissues tested,
supporting them being bona-fide editing sites ([Supplementary-material SD1-data], available via Dryad data repository).
Lastly, these sites show a clear sequence motif resembling the known ADAR motif:
depletion of G/C in 5' (only 8% and 12% of these sites have G and C in the
5’, respectively) and enrichment of G in 3' (41% of these sites have G in
the 3’).

Strong sites detection is based on a p-value calculated by the number of DNA reads. The
minimal number of DNA reads to achieve significance in this case is five DNA reads.

We did not apply another filter, which is the number of supporting RNA reads showing G.
Trinity decision to call the consensus site G was considered strong enough evidence for
G in the RNA reads. As explained above, Trinity might call a base even in the absence of
uniquely aligned, high quality reads, so the low number of reads supporting the
“G” call, or even the absence of any such reads, does not mean the site is
not an editing sites. Looking more closely at the sites with low RNA-reads coverage, we
find among our 5649 strong sites 5, 8 and 35 sites with 0, 1, 2 supporting RNA reads
showing “G”, respectively. These numbers should be compared to the number
of similar mismatches of the second most common modification type, which are 7(GA),
2(CT) and 5 (TC), respectively. We therefore agree that, despite our wish to minimize
parameters and avoid arbitrary cutoffs, requiring at least one (uniquely aligned, high
quality) RNA read to support editing is reasonable. We thus add this additional filter,
and remove the five strong sites from the dataset. Sites with even a single read showing
“G” are included (indeed, all of these eight sites, but only two out of
the five we excluded above, show evidence for editing in the additional three
nervous-system tissues tested).

The omission of these five sites (three recoding sites) has no visible effect on the
figures, and thus the figures need not be replaced. Numbers have been modified
throughout the paper, when needed, but the changes are always insignificant. Note that
the table of editing sites provides the full information on the number of DNA and RNA
reads, enabling the user to gauge the confidence of the specific site of interest.

We have added to the Methods section the following sentences: “We did not apply
additional read-number filters, as these seem to only marginally increase accuracy while
reducing the number of detected sites. However, we did exclude five strong sites for
which there were no uniquely-aligned, high-quality, supporting RNA reads. This brings
the number of strong A-to-G and non A-to-G modifications to 5,644 and 219, respectively.
The full list of weak and strong sites ([Supplementary-material SD1-data], available via Dryad data repository)
provides the number of DNA and RNA reads per site.”

*Did you apply filters to RNA and DNA reads? I mean reads with low quality and
positions at read ends*.

As mentioned in the Methods section, our alignments considered only uniquely aligned
reads and sites with quality-score Q≥30. However, Trinity de novo assembly takes
into account all reads and treats the quality score differently.

We did not remove read ends—our alignment was done to the transcriptome, and thus
the splicing-junction-related misalignments that are known to introduce errors at the
reads' ends are not expected. In addition, we used the local alignment
configuration of Bowtie2, which allows reads’ end “trimming” to
optimize alignment. Indeed, as shown in Figure 2—figure supplement 4, A-to-G mismatches are not overrepresented at
reads' ends.

The following text has been added to the Methods section: “In the following,
bases called with quality score Q<30 were discarded. Note, however, that Trinity
consensus sequence does take into account these bases, as well as reads that might have
not been uniquely aligned to the transcriptome. It is often customary to filter out
reads' ends when analyzing RNA-DNA mismatches. A main reason for that is the common
mismatches at reads' ends due to alignment artifacts when a splicing junction
occurs near the ends. In our case, as alignment is done to the transcriptome, we did not
observe any increase in AG mismatch rate near reads' ends (Figure 2—figure supplement 4), and thus no such filter was
used.”

*4) Have the authors considered the possibility that their results arise from
somatic genomic editing, rather than RNA editing? While for the human and macaque
control, the RNA and DNA samples are from the same tissues, in the case of squid, RNA
samples are from the tissues from the nervous system, while DNA is from the sperm
sack. To unequivocally conclude that the observations are indeed from RNA editing, I
guess that DNA and RNA need to be from the same biological source. Maybe the
investigation of the distribution of the relative proportion of reads supporting and
not supporting the edit could help here*.

As explained above, somatic mutations might indeed be a partial explanation of our false
discovery rate, but, as they should not be biased towards A-to-G (and certainly should
not introduce more A-to-G than T-to-C mismatches as they do not have strand preference),
their contribution is already accounted for in our estimated false discovery rate (based
on the non-AG mismatches), and they cannot account for a sizable fraction of our A-to-G
sites.

In principle, one might have speculated that an endogenous squid-specific DNA-editing
process leads to specific A-to-G mismatches at a level much higher than the random
mutations seen in other organisms. But: (i) we have no evidence for such a DNA deaminase
enzyme in the squid (or any other organism) while we do know ADAR enzymes are present;
(ii) even such a putative DNA deaminase is not expected to select the coding strand over
the other strand, so there is no explanation for the enrichment of A-to-G over
T-to-C.

In addition, we can exclude somatic mutations as a major contributor to our putative
editing sites list based on the following:

The same sites re-occur in different tissues and different animals (see Figure 2—figure supplement 2).

Randomly sampled sites were validated in additional different animals.

Sites appear in clusters, as expected for editing sites but not somatic mutations.

Nearby sites are not fully correlated (in the same read). Typically, for a pair of
adjacent sites we find reads showing all four combinations (A in both, G in both, A and
G, G and A), with only partial correlation. This is not expected for DNA-originated
mutations.

The sites show a clear sequence motif resembling the known ADAR motif (depletion of G/C
in 5'; enrichment of G in 3').

*5) Related to the above, the authors used RNA only from tissues from the nervous
system. Therefore, it is not possible to assess whether the phenomenon observed is
characteristic of this system, or it is actually systemic in the entire organism. I
think that sequencing RNA from some other non-nervous tissue could help to
distinguish between the two hypotheses*.

We thank the referees for this important comment. We have now checked the editing levels
in six additional non-nervous-system tissues at the sites already identified.
Interestingly, the editing level in these tissues is an order of magnitude lower than in
the five nervous system tissues previously studied. The data has been added to [Supplementary-material SD1-data]
(available via Dryad digital repository).

We have added to the main text the statement: “Consistently, editing levels
observed in non-nervous-system tissues are considerably lower ([Supplementary-material SD1-data])”.

We have also added to the Methods section information of the six additional tissues
tested: “Tissues were also dissected from non-neuronal regions: the branchial
heart, the Gill, the ventral epithelial layer on the pen, the marginal epithelial layer
on the pen, the iridophore layer of the skin, and the chromatophore layer of the skin.
Each of these six tissues originated from a different animal. RNA from all tissues was
extracted with the RNAqueous kit (Life Technologies), and genomic DNA was extracted from
the sperm sack using Genomic Tip Columns (Qiagen).”

And: “To characterize the modifications levels in non-neuronal tissues, Illumina
sequencing was again utilized to generate ∼23, ∼23, ∼19,
∼26, ∼19 and ∼14 million paired-end, 150 nt reads, using RNA from
the branchial heart, the Gill, the ventral epithelial layer on the pen, the marginal
epithelial layer on the pen, the iridophore layer of the skin, and the chromatophore
layer of the skin. The same alignment procedure was applied to quantify the modification
level at the previously described sites for each of the additional samples, revealing
considerably lower editing levels in the non-neuronal tissues ([Supplementary-material SD1-data],
available via Dryad data repository).”

*6) Regarding the characterization of RNA editing events, events tend to be
tissue specific. Are there events showing tissue specific levels? That is, cases in
which the gene locus in expressed at the same level in all tissues but editing levels
are different*.

[Supplementary-material SD1-data]
(available via Dryad digital repository) provides all data required to search for such
cases (A and G reads for each tissue). Indeed, one finds many cases in which the
expression level is similar but the editing level is very different. For example,
comparing OL and GFL tissues, there are 20,394 sites for which we obtained at least 100
reads per tissue, and the number of reads did not change more than 1.5-fold between
tissues. Interestingly, the average editing level in these sites is 1.5-fold higher in
the GFL tissue (9.4%) as compared to the OL tissue (6.1%). Looking for extreme
tissue-specificity among these similar-expression sites, we find 13 cases in which the
GFL sample exhibits editing levels >50% while the OL level at the same site are
<5%, but only 3 cases of the opposite scenario.

In summary, we do see tissue specificity even when expression levels are similar. The
data provided enables the reader to find such sites using his/her own parameters.

*7) It is a little bit disappointing that there is limited investigation in the
potential mechanisms behind the extensive editing observed. The authors could have at
least investigated ADAR with some additional detail. The RNA (and DNA sequence) helps
to delineate the ADAR sequence, and the RNA reads to estimate expression levels. Are
there multiple copies of ADAR in the squid genome? Is ADAR expressed at comparatively
higher expression levels than in organism with low editing levels (they can use the
mouse and human samples to make this comparison? Has the ADAR sequence in squid
diverged faster than expected? In specific domains? All these questions are quite
simple to answer*.

We agree that, in light of these findings, the structure and function of squid ADARs
could be very intriguing. Our de novo transcriptome identified three ADAR-like enzymes.
One which is similar to ADAR1, one similar to ADAR2, and a third one that we would
predict to be inactive because it contains mutations at key residues which are involved
in catalysis. Vertebrates also contain an inactive ADAR (ADAR3), however the
squid’s “inactive” ADAR appears equally similar to all three
vertebrate ADARs. The squid ADAR2 sequence has been well characterized and is the
subject of two previous publications by the Rosenthal lab. The squid ADAR1 sequence is
the subject of an ongoing investigation by the Rosenthal lab.

It is reasonable to hypothesize that high level of editing in squid is due to
comparatively high ADAR expression. However, it is difficult to compare the expression
levels in terms of reads per kilobase of exon per million mapped reads (RPKM) in the
absence of a full genome, because we can’t say whether unmapped reads are
actually mappable to poorly expressed, or poorly resolved, transcripts. That being said,
we did make a rough estimate of ADAR expression to answer this query. By ranking each
component from our transcriptome we found that squid ADAR1 ranks at 0.92 and squid ADAR2
ranks at 0.48 for the GFL tissue, where 1 is the most expressed component, and 0 the
least expressed component. These numbers are not very different from the situation in
the human brain, where (based on the normal total brain tissue in Human Body Map
dataset), ADAR1 is ranked at ∼0.9 and ADAR2 ∼0.5 (numbers are approximate,
as different exons behave differently, and identifying the expression levels of the
different splice variants is not trivial). This is only a rough estimate, and there are
several delicate problems in this comparison; therefore we prefer not to make the claim
that the expression levels are indeed similar in the text. However, we are working under
the hypothesis that other differences between squid and mammalian ADARs (see below) are
more likely to be of importance here.

Much is known about squid ADAR2 and it does indeed have several interesting features
that may explain, in part, high level editing. We’d like to stress, however, that
it is only part of the puzzle. Vertebrate ADAR2’s have an invariant domain
structure: they are composed of 2 N terminal double stranded RNA binding motifs (dsRBMs)
followed by a conserved catalytic domain. Squid ADAR2 can have an additional dsRBM at
its N terminus which is included in about half the transcripts through splicing
(Palavicini et al., 2009). Thus, there is a canonical ADAR2 with 2 dsRBMS and a
non-canonical one with three. When produced recombinantly in yeast and tested in vitro
on a squid K channel mRNA substrate, the non-canonical version edits far more sites.
This version also has a much higher affinity for dsRNA (Palavicini et al., 2012).
Another interesting feature of the squid ADAR2 is that its own messages are abundantly
edited, leading to multiple isoforms. The specificities of the individual isoforms have
yet to be tested.

We have also cloned and expressed Squid ADAR1. As with squid ADAR2, this enzyme has
notable differences when compared with its vertebrate counterparts. Vertebrate ADAR1s
are normally composed of three dsRBMs followed by a catalytic domain. Squid ADAR1 only
has 1 dsRBM followed by the catalytic domain. At its N-terminus, however, it contains a
highly basic domain that contains scores of phosphorylation sites. Squid ADAR1 messages
are also highly edited. A graduate student in the Rosenthal lab has produced recombinant
Squid ADAR1 and is currently characterizing its function for her doctoral project.
Because it contains many differences from vertebrate orthologs, a full-functional
characterization is a complex undertaking. Although its structure is clearly
interesting, we prefer not to include Squid ADAR1 sequence data in this manuscript
because only through structure-function studies can we assess whether its unique
features might contribute to high-level editing. We would like to assure the reviewer,
however, that these issues are very much on our minds and that we will be publishing
detailed accountings shortly.

We have added the following to the main text: “Have squid ADARs evolved novel
structure that account for the high-level editing? A past study showed that a squid
ADAR2 ortholog can be expressed as two isoforms due to alternative splicing: one, having
2 double-stranded RNA (dsRNA) binding motifs, resembles vertebrate ADAR2s. A second,
however, contains an “extra” dsRNA binding motif at its N-terminus. This
non-canonical isoform edits RNA more efficiently, edits more sites, and binds dsRNA with
a far higher affinity. Further Squid ADAR2 messages themselves contain many editing
sites, leading to many subtly different isoforms. An ADAR1 isoform is also present in
our transcriptome and is the focus of an ongoing study.”

*8) The authors also provide an adaptive explanation to the high levels of
editing observed in the squid genome, and hypothesize that, in contrast to current
assumptions, that extensive editing is common as a way to cope with temperature
adaption, except in mammals that, as homeotherms, would not require such a process.
This is, by the way, reminiscent of the isochore theory by Bernardi that would
separate homeotherm vertebrates from “cold-blooded” (poikilotherm)
vertebrates (to which, by the way, the authors may want to cite). If the authors were
correct that would indeed be a quite relevant result. They could easily employ their
pipeline in available vertebrate RNAseq data (for instance*, *http://www.sciencemag.org/content/338/6114/1587.full**)
to test this hypothesis.*

We thank the referees for this comment. The statement at the closing sentence of the
main text was indeed a bit too strong, and we would like to rephrase and clarify our
claim. We do not claim that *all* cold-blooded animals have extensive
recoding. Published data for *Drosophila*, the leaf-cutting ant
*Acromyrmex echinatior*, and *C. elegans* all show much
less recoding activity than we report for the squid. What we do intend to suggest is
that in those species where extensive editing does happen, it can be utilized for
temperature adaptation.

Thus, testing another cold-blooded species is not expected to help much, we presume most
species will not reproduce the exceptional level found in squid. We do intend to
explore, in future studies, the editing profile of species closely related to squid, in
order to better understand how this extensive phenomenon has evolved.

Parenthetically, the data in the reference suggested above cannot be used for our
purposes, as no matching DNA-reads are provided.

We have modified the closing sentence to read: “Most model organisms studied so
far are mammals which are homeotherms. Future studies of more diverse species are needed
to reveal the extent to which cold-blooded organisms might utilize extensive editing to
respond to temperature changes and other environmental variables”.